# Characteristics of clinician input in Canadian funding decisions for cancer drugs: a cross-sectional study based on CADTH reimbursement recommendations

Kristina Jenei  ,[1] Daniel E Meyers[2]

¹Department of Health Policy, The London School of Economics and Political Science, London, UK
²Department of Medicine, University of Calgary, Calgary, Alberta, Canada

**Correspondence to**
Ms Kristina Jenei;
k.jenei@lse.ac.uk

## ABSTRACT

**Objective** To examine characteristics of clinician input to the pan-Canadian Oncology Drug Review (pCODR) for cancer drug funding recommendations from 2016 to 2020.

**Design, setting and participants** Descriptive, cross-sectional study including 62 reimbursement decisions from pCODR from 2016 to 2020.

**Interventions** pCODR recommendations were analysed for the number of clinicians consulted on each submission, affiliation, number of submissions per clinician, declared financial conflicts of interest (FCOIs), randomisation, type of blinding, primary endpoint, study phase, and whether the study demonstrated improvement in overall survival (OS) and progression-free survival (PFS).

**Main outcome measures** The main outcome was clinician support for the initial funding recommendation. Secondary outcome measures were the association between clinician FCOIs and clinical benefit in positive recommendations.

**Results** The study consisted of 62 submissions, in which 48 included clinician input. A total of 129 unique clinicians provided 342 consultations. The majority (59%) provided input on less than 5 submissions; however, a small proportion (4%) consulted on over 10. Nearly all clinicians were physicians (125; 96%). From the 342 consultations, 228 declared financial conflicts (67%). The most common conflicts were payments for advisory roles (51%) and honorariums (23%). Of the 48 cancer drugs under review, clinicians recommended funding 46 (96%). Only 12 (25%) demonstrated substantial benefit, according to the European Society for Medical Oncology Magnitude of Clinical Benefit Scale score. Drugs recommended for funding were more likely to have improved PFS and OS data. However, most cancer drugs supported by clinicians demonstrated no change in health-related quality of life (HRQoL), including one that demonstrated worsened HRQoL. There was no statistically significant difference between FCOI status and recommending drugs with health gains.

**Conclusion** Clinicians offer crucial information on funding decisions. However, we found clinicians strongly supported funding nearly all cancer drugs under review, despite most not offering substantial benefit to patients nor gains in quality of life. While these drugs might be helpful options

---

## STRENGTHS AND LIMITATIONS OF THIS STUDY

⇒ The study examines data from the Canadian Agency for Drugs and Technology in Health reimbursement recommendations that are used to guide provincial funding decisions in Canadian provinces.

⇒ To the best of our knowledge, this study is the first to examine characteristics of clinician group input with the quality of evidence that underpin the drugs in Canadian funding decisions.

⇒ While the study uses the European Society for Medical Oncology Magnitude of Clinical Benefit Scale tool to assess clinical value, the cohort examined is a homogeneous sample of solid tumour indications, which cannot be generalised to all malignancies.

---

in clinical practice, funding numerous cancer drugs may be unsustainable for public health systems.

## INTRODUCTION

Country expenditure on cancer drugs has risen worldwide. Oncology medicines often account for the largest proportion of drug expenditure. Further, the costs of oncology service delivery have grown. In Europe, cancer care accounts for 30% of total hospital expenditure across Europe.[1] Indeed, sales of oncology medicines are expected to grow to $237 billion by 2024, with many new drugs under development.[2 3] In Canada, the largest proportion of Canadian public drug funding (22.9%) was spent on oncology agents.[4] Indeed, Canada is the third highest for total dollars spent on pharmaceuticals among 31 countries included in the Organisation for Economic Co-operation and Development.[5] The trend of rising costs is concerning due to uncertain or negligible health gain often offered by new therapeutics.[6] Furthermore, previous research has found weak correlation

between overall survival (OS) and the prices of new medicines at market launch.[7] Taken together, these challenges threaten the sustainability of publicly funded healthcare systems globally.[8–10]

Many countries with publicly funded healthcare systems employ health technology assessments (HTAs) to evaluate the value of new health technologies. These assessments often include stakeholder perspectives to ensure value perspectives are considered in the funding decision. In Canada, the Canadian Agency for Drugs and Technology in Health (CADTH) is the national HTA body that guides funding recommendations for Canadian provinces. CADTH, and its oncology assessment arm, the pan-Canadian Oncology Drug Review (pCODR),[11] evaluate medicines through a deliberative framework that includes clinical and economic evidence, clinician and patient feedback.[12] While there are special authorisations, generally only medicines that are funded by the province are accessible to patients. Once a drug is reviewed, the pCODR Expert Review Committee (pERC)[13]—an appointed interdisciplinary body comprised of clinicians, ethicists, health economists and patients— provides the final reimbursement recommendation. While each Canadian province independently decides whether a drug will be funded under their public drug plans, provinces often follow CADTH recommendations.[14] Since 2021, the pCODR and Common Drug Review (HTA assessment all other health interventions) have been aligned within one process. However, the responsibility of recommendations to pERC and the process for clinician group feedback remains the same.

Clinicians are involved in Canadian cancer drug funding decisions through two mechanisms. First, every drug review team includes oncologists to develop the systematic review protocol, interpret and provide real-world applicability for the study results. Once pERC reviews the clinical and economic reports and provides an initial funding recommendation, there is a second opportunity for clinician feedback. Since October 2016, CADTH provides an opportunity for additional feedback through external 'calls for clinician input'. Registered clinicians, such as oncologists, pharmacists and oncology nurses, can submit feedback. While submissions can be made individually, CADTH encourages clinicians to submit in a group or healthcare association. Each submission must include an oncologist. This feedback is used to determine the final funding recommendation and published on the pCODR website.

Given trial participants are often non-representative of clinical practice, clinician and patient perspectives are important to prioritise understanding the value of medicine in real-world settings. However, previous research exploring the role of clinicians in Canadian funding decisions found widespread financial conflicts of interest (FCOIs).[15] In the USA, FCOIs are widespread in oncology[16] and have been found to influence prescribing patterns, formulary recommendations, research and recommendations for clinical practice guidelines.[17] Given these associations, we sought to evaluate characteristics of clinician input beyond FCOIs, including the

alignment between funding support and underlying clinical evidence for the cancer drugs under review.

## METHODS

We conducted a cross-sectional, descriptive study of reimbursement recommendations to pCODR for solid tumour indications between 2016 and 2020. We adhered to the Strengthening the Reporting of Observational studies in Epidemiology reporting guidelines.

### Source of data

A search of the pCODR website was completed to identify characteristics of physician submissions and supporting evidence. We used a sample of solid tumour indications from an existing dataset.[6] Supportive care medicines, haematological neoplasms, paediatric indications and biosimilars were excluded. By focusing on solid tumour indications, the dataset was homogeneous which allowed us to assess the clinical benefit using the European Society for Medical Oncology Magnitude of Clinical Benefit Scale (ESMO-MCBS),[18] which was designed for solid tumour indications. We extracted the name of the drug, date and year of reimbursement recommendation and supporting clinical trial information, such as randomisation, blinding, OS and progression-free survival (PFS) outcomes, health-related quality of life (HRQoL) data and estimated price (per 28-day cycle). While final drug prices are confidentially negotiated with the manufacturer after the funding recommendation, estimated prices are included in the pCODR economic evaluation reports. For more information on how these data were collected and how the ESMO-MCBS scores were calculated, see the previous publication.[6]

### Clinician input on cancer drug submissions

In addition to the clinical evidence supporting each drug, we extracted data related to the clinician submissions. These data included the number of clinicians consulted on each submission, affiliation, number of submissions per clinician and declared FCOIs. Clinicians provide feedback on the initial funding recommendation. On the submission forms, clinicians are asked whether they 'agree', 'disagree' or 'agree in part' with the initial funding recommendation. With this feedback, pCODR can revise the final funding recommendation. While there are often numerous clinicians involved in each drug review, the feedback form is submitted collectively. When the clinicians agreed with a positive funding recommendation or disagreed with a negative funding recommendation, we coded this as 'fund'. When the clinicians agreed with a negative recommendation or disagreed with a positive funding recommendation, we coded this as 'do not fund'. When the clinician feedback was to 'agree in part', we reviewed comments in the feedback documents to ensure coding was accurate.

### Conflict of interest data

Despite one collective feedback form, FCOI forms are submitted individually. FCOI forms require clinicians to

disclose whether they have received payments over the previous 2 years from any company or organisation that may have direct or indirect interest in the drug under review. The forms require disclosure on the type of activity the payment was for (eg, advisory board, conferences, honoraria, research, sponsorship of events, operations, travel, among others). Clinicians must also provide the names of the manufacturers who they have received funds from, including whether these payments exceeded $C10 000 and whether they have personal or commercial relationships with any interest group in relation to the drug under submission. Information related to all these categories was also extracted. We then investigated the association between FCOI status (conflict with submission, conflict with other companies, no conflict) and clinical benefit for drugs that received positive recommendations to understand whether FCOI would impact funding recommendations.

### Data analysis

Descriptive data and counts are presented throughout this manuscript. While FCOI documentation is submitted individually, clinician feedback is sought through groups and cannot be disaggregated further. Therefore, due to CADTH procedures, we analysed clinician group submissions. FCOIs were analysed as individuals given available data. We explored the association of FCOI status (conflict with the company, conflict with another company, no conflict) in recommending drugs with meaningful benefit (defined as improvements in ESMO-MCBS score, OS benefit and HRQoL status). The purpose of this analysis was to understand the role of FCOIs in recommending drugs with or without health gains. Data per submission were aggregated to compare these factors. Both analyses used Fisher's exact test. Data analysis was conducted using Microsoft Excel (V.16.73) and R Statistical Software (V.2023.06.1+524).

### Patient and public involvement

No patients or members of the public were involved.

### RESULTS

Table 1 reports the descriptive characteristics of clinicians who were consulted during cancer drug funding decisions. Overall, there were 62 drug funding reviews by CADTH for solid tumour indications between 2016 and 2020. Of these, clinician group input was available for 48 submissions. For 14, there was no clinician feedback published on the pCODR website (there are no reasons provided for why these submissions did not have clinician input). On average, 4 (IQR: 2–8) clinicians provided feedback per drug. In the 48 drug reviews, there were 342 submissions from clinicians for feedback on funding recommendations. However, there were only 129 unique clinicians, which suggests clinicians often consulted on more than one drug. Of these, 124 (96%) were physicians and 5 (4%) were oncology pharmacists. Most clinicians

**Table 1** Characteristics of clinicians who submitted input to the pan-Canadian Oncology Drug Review

| Characteristics of individual clinicians consulted | |
| --- | --- |
| Total number of clinician feedback submissions | 342 (100) |
| Unique clinicians | 129 (100) |
| Number of funding decisions consulted | |
| 1 | 76 (59) |
| 2 | 16 (12) |
| 3–10 | 33 (26) |
| More than 10 | 4 (3) |
| Affiliation | |
| Physician | 124 (96) |
| Oncology pharmacist | 5 (4) |
| **Characteristics of clinician group submissions** | |
| Reimbursement decisions (2016–2020) | 62 (100) |
| Reimbursement decisions with clinician submissions* | 48 (100) |
| Clinician recommendation | |
| Fund | 46 (96) |
| Do not fund | 2 (4) |
| Median number of clinicians who offered feedback per drug review (IQR) | 4 (2–8) |

*Study sample.

(76; 59%) consulted on only one drug review. However, a small minority (four; 3%) made repeat submissions on more than 10 drugs.

Table 2 reports the clinician recommendations and supporting evidence. Of the 48 submissions that included clinician input, 46 (96%) had positive clinician feedback to fund the drug. This included 12 (26%) drugs that received negative recommendations from CADTH in which clinicians submitted feedback in support of funding the drug. Only two (4%) submissions believed the drug should not be funded.

Of the drugs recommended for funding, most (32; 70%) did not offer substantial benefit, as per ESMO-MCBS.

In terms of clinical benefit, clinicians recommended funding 18 (39%) drugs that had data that demonstrated an OS benefit, 12 (26%) without an OS benefit and 16 (35%) that had no data on OS. Similarly, 25 drugs (54%) that demonstrated an improved PFS benefit, 5 (11%) that did not improve PFS and 16 drugs (35%) that did not have any data related to PFS received funding support from clinicians. Nearly two-thirds of drugs (32; 70%) that received funding support by clinicians demonstrated no change in HRQoL, with one that worsened HRQoL outcomes. Most cancer drugs that received clinician funding support were between $5000 and $9999 per month (59%), including 14 (30%) that were over $10 000 per 28-day cycle (table 3).

**Table 2** Clinician group funding recommendation and supporting evidence

| | Clinician group recommendation (n=48) | |
|---|---|---|
| | Fund (n=46) | Do not fund (n=2) |
| **CADTH decision** | | |
| Positive | 29 (63) | 0 (0) |
| Negative | 12 (26) | 2 (100) |
| **Submission with conflicts** | | |
| Yes | 44 (96) | 1 (50) |
| No | 2 (4) | 1 (50) |
| **Submissions with conflicts with manufacturer** | | |
| Yes | 34 (74) | 1 (50) |
| No | 12 (26) | 1 (50) |
| **ESMO-MCBS score** | | |
| Substantial benefit | 23 (50) | 2 (100) |
| No substantial benefit | 23 (50) | 0 (0) |
| **RCT** | | |
| Yes | 34 (74) | 2 (100) |
| No | 12 (26) | 0 (0) |
| **Phase** | | |
| 1 | 3 (7) | 0 (0) |
| 2 | 10 (22) | 0 (0) |
| 3 | 33 (72) | 2 (100) |
| **Blinded studies** | | |
| Yes | 20 (43) | 2 (100) |
| No | 14 (30) | 0 (0) |
| NR | 12 (26) | 0 (0) |
| **OS** | | |
| Improved | 18 (39) | 0 (0) |
| Not improved | 12 (26) | 1 (50) |
| NA | 16 (35) | 1 (50) |
| **PFS data** | | |
| Improved | 25 (54) | 0 (0) |
| Not improved | 5 (11) | 0 (0) |
| NA | 16 (35) | 2 (100) |
| **HRQoL** | | |
| Improved | 5 (11) | 0 (0) |
| No change | 32 (70) | 2 (100) |
| Worsen | 1 (2) | 0 (0) |
| NA | 5 (11) | 0 (0) |
| **Price (per 28 cycle)** | | |
| Under $5000 | 5 (11) | 0 (0) |
| $5000–$9999 | 27 (59) | 1 (50) |

Continued

**Table 2** Continued

| | Clinician group recommendation (n=48) | |
|---|---|---|
| | Fund (n=46) | Do not fund (n=2) |
| Over $10 000 | 14 (30) | 1 (50) |

CADTH, Canadian Agency for Drugs and Technology in Health; ESMO-MCBS, European Society for Medical Oncology Magnitude of Clinical Benefit Scale; HRQoL, health-related quality of life; NA, not available; NR, not reported; OS, overall survival; PFS, progression-free survival; RCT, randomised controlled trial.

We found numerous FCOIs (figure 1). Of the 48 submissions made by clinician groups, 46 that had complete FCOI disclosures that were able to be analysed (9 of 342 individual documents; 3%) were incomplete. Of the 342 instances where clinicians were consulted during the drug review, 228 (67%) reported FCOIs (table 3). Of these, 149 (44%) were with the manufacturer of the drug under review. The average number of conflicts a clinician disclosed was 3 (IQR: 2–6). Advisory roles were the most common type of conflict (166; 51%), followed by honorariums (23%), payments to attend conferences, research and sponsorship of events (7%, respectively). We found nine documents (3%) that were incomplete.

We extended this analysis to understand the role of FCOIs in recommending drugs with or without meaningful clinical benefit in positive clinician group recommendations (table 4). We found that there was no

**Table 3** Financial conflicts of interest among clinicians consulted on drug funding decisions

| Characteristic | Submissions (n=342) |
|---|---|
| Clinician submissions declaring conflicts | 228 (67) |
| Clinician submissions with conflicts with the manufacturer of the drug under review | 149 (44) |
| Average number of conflicts per submission (IQR) | 3 (2–6) |
| Types of conflicts reported | |
| Advisory role | 166 (51) |
| Honoraria | 74 (23) |
| Conference | 22 (7) |
| Research | 22 (7) |
| Events | 23 (7) |
| Operations | 2 (1) |
| Travel | 3 (1) |
| Speaking | 12 (1) |
| Incomplete conflict of interest documentation | 9 (3) |

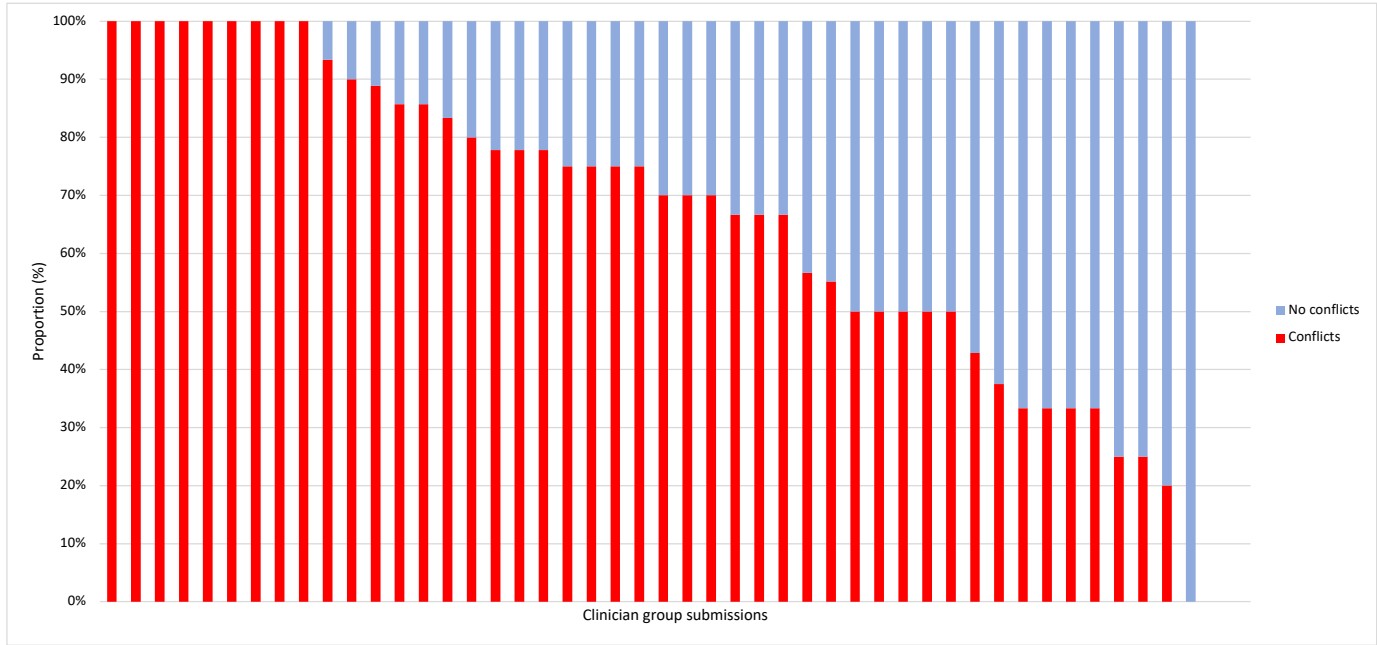

**Figure 1** Proportion of clinicians reporting conflicts per individual submission (n=46). [a]Two submissions were removed from the figure due to incomplete information.

statistically significant difference between clinician FCOI status and clinical benefit per ESMO-MCBS score, OS benefit, nor improvements with HRQoL.

## DISCUSSION

We sought to examine characteristics of clinicians consulted in Canadian cancer drug funding decisions from 2016 to 2020. We examined clinician recommendations and supporting evidence for the cancer drugs under review for reimbursement. Further, we explored the extent of FCOIs disclosed in clinician submissions and their association with recommending drugs with

meaningful clinical benefit. We found that clinicians who consult on Canadian funding decisions support funding the cancer drug under review. Of the 48 drugs, 46 (96%) received supportive funding feedback from clinicians, even when most drugs did not offer substantial benefit to patients nor improved quality of life. These findings have important implications.

First, our findings highlight a substantial proportion of new cancer medicines are introduced in the Canadian market with high prices and uncertain or negligible value. It is reasonable to assume many clinicians would prefer robust, mature evidence demonstrating improvements in

**Table 4** Association between the number of clinicians declaring financial conflicts of interest (FCOIs) and supporting evidence for positive group recommendations*

| Clinical gains | Conflict with submission | Conflict with another company | No conflict | P value |
|---|---|---|---|---|
| ESMO-MCBS score | | | | |
| Substantial benefit | 84 | 34 | 45 | 0.1 |
| No substantial benefit | 65 | 34 | 60 | |
| OS benefit | | | | |
| Improved | 56 | 31 | 38 | 0.29 |
| Not improved | 40 | 12 | 25 | |
| HRQoL status | | | | |
| Improved | 26 | 12 | 22 | 0.77 |
| Worsened | 12 | 4 | 12 | |

*FCOI status is defined as the number of clinicians declaring an FCOI that participated in the drug submission process; positive clinician funding recommendations include partial agreement with positive CADTH recommendation; total values differ between supporting evidence categories due to data availability of the clinical gain; association between FCOI status and clinical gains was analysed using Fisher's exact test.

CADTH, Canadian Agency for Drugs and Technology in Health; ESMO-MCBS, European Society for Medical Oncology Magnitude of Clinical Benefit Scale; HRQoL, health-related quality of life; OS, overall survival.

patient-centred outcomes, such as improvements in OS and quality of life. However, in a previous study, we found that only half of cancer drugs that received a positive funding recommendation from pCODR had evidence of improved survival.[6] Similarly, the Patented Medicine Prices Review Board—the federal agency responsible for regulating drug prices of new patented medicines in Canada—classified over 90% of new medicines as offering 'moderate, little or no improvement' in clinical benefit.[19] Therefore, in this ecosystem, objective and clear decision-making is required. Influence from conflicts or vested interests might add strain to health budgets.

Second, our findings might highlight conflicting principal–agent relationships between national HTA bodies, pharmaceutical manufacturers and clinicians. The duty of a clinician to care for individual patients differs from HTA bodies who attempt to maximise population health outcomes for every dollar spent. For clinicians, it might be advantageous to have numerous treatment options to offer patients, including a spectrum of available medicines given differential responses to individual therapies. However, at the national reimbursement level, funding numerous high-priced cancer drugs incurs a large budget impact. Indeed, 27 drugs (59%) that received positive funding support from clinicians were priced between $5000 and $9999 per month. This is concerning in oncology where many high-priced cancer drugs approved through regulatory agencies often only offer marginal benefit, if any at all.[6 20 21]

Third, we found numerous conflicts of interest. Of the 342 submissions, 228 (67%) clinicians declared conflicts and nearly half were directly with the manufacturer of the drug under review. Clinicians who declared FCOIs with the manufacturer of the submission were more likely to recommend funding the drug. However, there was no statistically significant difference between the distribution of individual clinician FCOIs and recommendation for drugs with meaningful clinical benefit. This is an interesting finding that warrants further investigation to understand whether a true cause exists but may be related to the previously discussed desire to have numerous options for patients given treatment responses that may differ from the clinical trial results. Given the overwhelming support for nearly all drugs in our sample, this finding may be skewed. Furthermore, future research might repeat this study with the inclusion of haematological malignancies given the recently released ESMO-MCBS for this population.[22] Our results are aligned with previous Canadian research. Lexchin found that 75% of clinicians who disagreed with a negative funding recommendation declared a conflict with the manufacturer under review.[15] However, in the nine negative initial funding recommendations, pCODR changed their final recommendation once. Similarly, in our study, the initial funding recommendation was only changed once after stakeholder feedback. Furthermore, we found 9 of 342 (3%) FCOI documents that were incomplete, which raises concerns about the disclosure process. The

literature about FCOIs and clinician behaviour suggests that there is a relationship between financial payments to clinicians and prescribing patterns, formulary recommendations, research and recommendations for clinical practice guidelines.[16 17]

Our findings raise concerns about the use of stakeholder perspectives in funding decisions and are relevant to other countries outside of Canada that use similar HTA methodology. Countries, such as England, Italy, New Zealand, Australia, among others, include input from a variety of external groups in funding decisions.[23] However, Canada is the only country that includes patient and clinician group feedback in every decision. Other countries include stakeholder input as deemed necessary, such as funding a drug for a rare or severe diseases. While CADTH includes patient and clinician group feedback to ensure fair decisions, the funding recommendations from these groups are complicated by numerous FCOIs and overwhelmingly positive support for the drug. Furthermore, we found several clinicians who consult on decisions more than once, with a small proportion consulting on over 10. Repeat consultations may be warranted by the nature of oncology, where novel therapeutics treat small, specialised populations, similar to those of rare diseases. To manage provider-induced demand, CADTH may want to reconsider how the agency engages with clinicians, or whether the agency ought to incorporate external stakeholder feedback on every funding decision, given overwhelming support for funding nearly every drug independent of clinical benefit. Furthermore, CADTH, among other HTA bodies, may want to be more explicit for how stakeholder input is used in funding decisions.

### Strengths and limitations

Our retrospective cohort study has strengths and limitations. It is unique as it is the first evaluation of characteristics of clinician input in Canadian funding decision that includes an assessment of the supporting evidence of the drugs under review. However, we encountered limitations. First, we were limited to a cohort of cancer drugs that was accepted and reviewed by CADTH. Second, feedback forms where clinicians indicate their support for drug funding are submitted collectively, not individually. Therefore, clinician input cannot be attributed to individual views. However, we were able to examine FCOI status (submitted individually) and the clinical benefit status of the recommended drug. Third, we are limited by a cohort of submissions for solid tumour indications which might differ compared with other malignancies. However, this approach allows for homogeneous dataset for which clinical benefit can be measured using the ESMO-MCBS. Further, several methodological limitations have been discussed regarding the ESMO-MCBS tool[24] (use of the lower limit CI, focus on primary endpoints even for studies that use unvalidated surrogate endpoints, use of quality of life data and no weighted arguments for values) which should be acknowledged given it is the measure of value within our study. Lastly, we are limited

by the publicly available data in pCODR reports. Given the complexity of evaluating evidence from oncology clinical trials, these reports might not offer insight into the nuanced discussions that likely occurred throughout the pCODR deliberations. Future research might build on our study to further examine the effects of FCOIs on the CADTH decision with more sophisticated statistical methods or qualitative analysis.

## CONCLUSIONS

Clinicians offer crucial information on funding decisions. However, we found overwhelming support for nearly all drugs under review, even for medicines that did not demonstrate improved OS benefit nor quality of life—two of the most important patient-centred outcomes. Further, we found numerous FCOIs. These findings raise questions about the role of stakeholder input in HTA funding decisions and possible incongruity between objectives of national HTA bodies and practising clinicians.

**Acknowledgements** Kristina Jenei is supported by a Canadian Institutes of Health Research Doctoral Foreign Study Award (181603)

**Contributors** KJ accepts full responsibility for the work and conduct of the study, had access to the data and controlled the decision to publish. DEM was involved in the interpretation, drafting and editing of the manuscript.

**Funding** The authors have not declared a specific grant for this research from any funding agency in the public, commercial or not-for-profit sectors.

**Competing interests** None declared.

**Patient and public involvement** Patients and/or the public were not involved in the design, or conduct, or reporting, or dissemination plans of this research.

**Patient consent for publication** Not required.

**Ethics approval** The study examined publicly available data and therefore was exempt from institutional review board approval, in accordance with the US Department of Health and Human Services (45 CFR §46.102(f)).

**Provenance and peer review** Not commissioned; externally peer reviewed.

**Data availability statement** Data are available upon reasonable request.

**ORCID iD**
Kristina Jenei http://orcid.org/0000-0002-3635-5212

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
