## [Reviewer comments · BMJ Open]

ARTICLE DETAILS

TITLE (PROVISIONAL)	Characteristics of clinician input in Canadian funding decisions for cancer drugs: a cross-sectional study based on CADTH reimbursement recommendations
AUTHORS	Jenei, Kristina; Meyers, Daniel E.

VERSION 1 – REVIEW

REVIEWER	Phelps, Charles University of Rochester
REVIEW RETURNED	12-Aug-2022

GENERAL COMMENTS	This is an interesting analysis. It illuminates several key issues about the interaction of pharmaceutical companies, clinicians and regulatory agencies. But it could be a lot better. The focal point of this study are clinicians, many of whom have dual roles in this setting. In economic theory, we would say that they have conflicting principal-agent relationships with both the pharmaceutical companies and pCODR, the regulatory agency. We must remember that COIs are, at their heart, situations, not behavior. The question is, what behavior emerges in these conflicted situations? Most of the drugs recommended for approval had either no gain in HRQoL or even a worsening. This highlights overall survival (OS) and Progression-Free Survival (PFS). Of the total, 16 of the drugs had "NA" for improvement in OS or PFS, and apparently only 25 that offered some survival benefit. These differences in drug benefit are a key to what else can be learned from these data. (As an aside, I can't understand how more drugs has some PFS than OS, since OS should have a higher total, i.e., those drugs that improved survival, but not in a progression-free manner). Since the clinicians almost uniformly recommended funding the drugs, it's impossible to assess the role of conflict of interest by analyzing approve/don't approve choices. We have to turn instead to the more-specific recommendations such as funding drugs that have no benefit. This "no benefit" result could be measured in two different ways. First, they could of course use the ESMO-MCBS measure, but, after reading the manual about it on their web site, it seems that this is yet again one of these arbitrary measures of benefit that oncologists seem fond of cooking up, with no significant underpinning of theory about how to combine gains in LE and HRQoL.
---

	The second way would be to classify drugs in one of four states: (1) Benefits both survival (using hazard ratios or whatever) AND HRQoL; (2) benefits survival, but not HRQoL; (3) benefits HRQoL but not survival; (4) Benefits neither. Similarly, the COIs can be categorized in a loose sequence. First, conflicts with the actual manufacturer are more important than other conflicts. Then, I'd argue, advisory roles are more important than honoraria, conference attendance, etc. The authors may be able to rank the importance of these COIs better than I can. The analysis would then explore the role of financial conflicts on the various types of drugs, focusing, I'd guess, on the 4th category perhaps – recommending funding when no apparent benefit of any kind existed. I would strongly recommend that the analysis be carried out at the individual recommendation level, not the level of the drug. Table 2 shows the attributes of all 342 individual submissions, but table 3, for reasons I don't understand, has only 48 "recommendations," which (I gather) are the combined recommendation from all physicians responding on each drug. I also think that this analysis could benefit from a multiple regression structure, using models such as logit or probit for the yes/no recommendation choice.
--	--

REVIEWER	Godman, Brian University of Strathclyde
REVIEW RETURNED	24-Aug-2022

GENERAL COMMENTS	Thank you for this very interesting paper in which you raise a number of relevant points - not least the continued request for high prices for new cancer medicines for solid tumours despite limited health gain (OS and HRQOL - Table 3). This practice has been continuing for some time, e.g. the cost of cancer care also now accounts for up to 30% of total hospital expenditure across Europe (Simoens S et al. What Happens when the Cost of Cancer Care Becomes Unsustainable. European Oncology & Haematology. 2017;13:108-13), with sales of oncology medicines are expected to grow to \$237 billion by 2024 and further with multiple companies researching in this area (Waters R, Urquhart L. EvaluatePharma® World Preview 2019, Outlook to 2024. 2019. - https://info.evaluate.com/rs/607-YGS-364/images/EvaluatePharma_World_Preview_2019.pdf and IMS Institute for Healthcare Informatics. Global Oncology Trend Report - https://www.scribd.com/document/323179495/IMSH-Institute-Global-Oncology-Trend-2015-2020-Report). This is helped by the emotive nature of the disease area (Haycox A. Why Cancer? Pharmacoeconomics. 2016;34:625-7). As a result - see new oncology medicines being launched at high prices with limited health gain - challenging the sustainability of healthcare systems providing universal access for their citizens - discussed in e.g. Cohen D. Cancer drugs: high price, uncertain value. Bmj. 2017;359:j4543 and Godman B et al. Potential approaches for the pricing of cancer medicines across Europe to enhance the sustainability of healthcare systems and the implications. Expert Rev Pharmacoecon Outcomes Res. 2021;21:527-40. I have some minor suggestions to make to enhance interest/ flow in the paper. These include:
---

	a) Introduction - I would move comments on Page 13 lines 33 - 44 plus some of the above to state why Cancer for this paper. The comments you make about Canada are particularly important for other similar healthcare systems - especially the rising costs of cancer medicines in the public fund (which will increase with most medicines to treat patients in ambulatory care now available as low cost generics/ biosimilars) b) Comments about the value of OS and PFS in solid tumours. There are concerns with how reliable PFS is in solid tumours as a reliable outcome measure especially when resources are constrained (e.g. Paoletti X et al. Assessment of Progression-Free Survival as a Surrogate End Point of Overall Survival in First-Line Treatment of Ovarian Cancer: A Systematic Review and Meta-analysis. JAMA Netw Open. 2020;3:e1918939; Prasad V et al. The Strength of Association Between Surrogate End Points and Survival in Oncology: A Systematic Review of Trial-Level Meta-analyses. JAMA Intern Med. 2015;175:1389-98 and Cortazar P et al. Pathological complete response and long-term clinical benefit in breast cancer: the CTNeoBC pooled analysis. Lancet. 2014;384:164-72 to name just a few). In fact when clinicians, health authorities and pharmacists in the UK established criteria for paying more for new cancer medicines when the purchaser/ provider split was established in the UK - only really concentrated on OS (minimum of 3 to 6 months additional survival) - Ferguson JS et al. New treatments for advanced cancer: an approach to prioritization. Br J Cancer. 2000;83:1268-73. In addition - concerns with some of the leniency of the ESMO criteria among European payers and their advisers - especially regarding the high weighting given to PFS - resulted in the publication of their concerns - Wild C et al. Utilisation of the ESMO-MCBS in practice of HTA. Ann Oncol. 2016;27:2134-6. However - overall endorse the use of ESMO criteria in this paper It may also be worth commenting that we are seeing an appreciable number of new cancer medicines being launch with limited data - but still high requested prices (discussed in e.g. Godman et al). Hence do need objective decision making - not unduly influenced by clinicians with a vested interest promoting new cancer medicines at high prices even with limited/ no health gain vs. current standards
--	--

REVIEWER	Yu, Genghua Central South University
REVIEW RETURNED	27-Oct-2022

GENERAL COMMENTS	This paper examine characteristics of clinician input to the pan-Canadian Oncology Drug Review (pCODR) for cancer drug funding recommendations from 2016 to 2020. The motivation for this paper is interesting. It appears to be a survey and statistical report. I have some questions:  1. The authors think that some supported funds may not offer substantial benefits but do not go further into analyzing reasons for the results and how to support them selectively. 2. The contribution of the paper is not very clear. How should this information used for fund recommendation be judged for reliability? What information is more helpful in evaluating the potential benefit of funding?
---

REVIEWER	Stewart, Roy Rijksuniversiteit Groningen, Department of Health Sciences, Community and Occupational Medicine, University Medical Center Groningen
REVIEW RETURNED	29-Mar-2023

GENERAL COMMENTS	In the article entitled "Association of clinician input and quality of evidence supporting cancer drug funding decisions in Canada" (bmjopen-2022-066378) the authors formulated that they sought to evaluate characteristics of clinician input beyond financial conflict of interests (FCOIs), including the association with funding support and underlying clinical evidence for the cancer drugs under review. The main outcome is clinician support for the initial funding recommendation. The paper is fairly well written and primarily written for the Canadian health system. Major Revision: In the strengths and limitations the authors claim that the study is the first to examine the association between clinician funding recommendations and the evidence underpinning cancer drug reimbursement decisions. This claim should be clearly substantiated and argued with respect to Lexchin's publication (3) and Meyers' two publications (1,2) given the period under investigation and the variables used to justify this claim. Table 5 of Lexchin's article also examines an association. In Lexchin's article this association is analysed using a Fisher exact test, which test was not used in the bmjopen-2022-066378, where the associations are only presented descriptively. So the addition of, say, a Fisher exact test or other tests would already be an improvement to the associations presented in table 3 (page 11). Other research techniques that could be applied in this manuscript under review are network analysis, but multilevel analysis could also be a useful analysis technique to analyse the information currently scattered in Tables 1, 2 and 3 into a single whole. This allows covariates to be taken into account in a multilevel analysis. The figure on page 18 could be better presented by placing the 100% conflict submission numbers together. The submission numbers have no meaning to the reader. Moreover, the numbers 29 and 35 are disturbing and should be dropped from the figure and can be listed at the bottom of the table as has been done now. Minor Revision: What is the meaning of the (f) in line 11 on page 7
---

REVIEWER	Chaiyasong, Surasak International Health Policy Program, Health Promotion Policy Unit
REVIEW RETURNED	11-Apr-2023

GENERAL COMMENTS	This study entitled "Association of clinician input and quality of evidence supporting cancer drug funding decisions in Canada", aimed to examine characteristics of clinician input to the pan-Canadian Oncology Drug Review for cancer drug funding recommendations from 2016 to 2020. The study does not use any inferential statistics to examine associations between clinician funding recommendation and supporting evidence. Only descriptive statistics are presented. It seems to be difficult to compare proportions between two groups due to that a number of clinician recommendation with "do not fund" is too small (n=2); any associations may be found by chance. Although Chi-square test is not applicable, Fisher exact test is suggested to use for examining associations between the variables.
---

	Moreover, there is no analysis section in this study. It should be described and informed audience how this study treated and analyzed the data in particular to examine the associations. Minor points. In tables, numbers and percentages of the study samples should be provided in all cells and rechecked. Study design is defined as a retrospective cohort study. I think that it can be named a retrospective study but do not think that a cohort study is applicable. However, if a terms cohort study is used, some epidemiological terms should be clearly defined i.e. exposure and non-exposure, outcomes measurement, incidence and follow-up time.
--	--

VERSION 1 – AUTHOR RESPONSE

Reviewer	Comment	Response	Location of change
Associate editor	- There were no inferential statistics used (just descriptive), so I am not sure one can talk of an “association between clinician drug funding recommendations and supporting evidence for the cancer drugs under review”. So this needs to be toned down.	Thank you for the comment. We have removed all wording regarding “association”	Throughout manuscript
	- Moreover, I find the discussion section very editorialized. You can see the point the authors are trying to make but that is not backed up by a robust analysis. They should just stick with the descriptive findings and not draw very definitive conclusions, which seems to be the case.	Thank you. The discussion has been adapted.	Page 10-11
Editor	- Please revise the title of your manuscript to include the research question, study design and setting. This is the preferred format of the journal.	Thank you. We have adapted the title “Characteristics of clinician input in Canadian funding decisions for cancer drugs: a cross-sectional study based on CADTH reimbursement recommendations”	Page 1
	- Please revise the ‘Strengths and limitations of this study’ section of your manuscript (after the abstract). This section should contain up to five short	Strengths and limitations of the study are revised to relate to the methods.	Page 3

	bullet points, no longer than one sentence each, that relate specifically to the methods. The novelty, aims, results or expected impact of the study should not be summarised here.		
	- Please include, as a supplementary file, the precise, full search strategy (or strategies) for all databases, registers and websites, including any filters and limits used.	The full search strategy is described in the methods, including a citation to a previous manuscript that outlines the data collection in full. "For more information on how these data were collected and how the ESMO-MCBS scores were calculated, see the previous publication.¹⁷ "	Page 7

Reviewer 1	This is an interesting analysis. It illuminates several key issues about the interaction of pharmaceutical companies, clinicians and regulatory agencies. But it could be a lot better. The focal point of this study are clinicians, many of whom have dual roles in this setting. In economic theory, we would say that they have conflicting principal-agent relationships with both the pharmaceutical companies and pCODR, the regulatory agency. We must remember that COIs are, at their heart, situations, not behavior. The question is, what behavior emerges in these conflicted situations? Most of the drugs recommended for approval had either no gain in HRQoL or even a worsening. This highlights overall survival (OS) and Progression-Free Survival (PFS). Of the total, 16 of the drugs had "NA" for improvement in OS or PFS, and apparently only 25 that offered some survival benefit. These differences in drug benefit are a key to what else can be learned from these data. (As an aside, I can't understand how more drugs has some PFS than OS, since OS should have a higher total, i.e., those drugs that improved survival, but not in a progression-free manner). Since the clinicians almost uniformly recommended funding the drugs, it's impossible to assess the role of conflict of interest by analyzing approve/don't approve choices. We have to turn instead to the more-specific recommendations such as funding drugs that have no benefit. This "no benefit" result could be measured in two different ways. First, they could of course use the ESMO-MCBS measure, but, after reading the manual about it on their web site, it seems that this is yet again one of these arbitrary measures of benefit that oncologists seem fond of cooking up, with no significant underpinning of theory	 • Thank you for this reviewers' thoughtful comments and suggestions. We agree that there are conflicting principal-agent relationships with the pharmaceutical companies and pCODR. We have added this language to the discussion where we discuss a "mismatch" of objectives. Discussion (page 10) third paragraph. • Further, we added an additional paragraph at the end of the Discussion (page 11) that provides an overview for how our findings might be interpreted in other countries that use similar HTA methods as Canada. • In terms of OS and PFS, more drugs (25) submitted for funding recommendations had PFS benefit over OS benefit (16). This is because many drugs are submitted earlier than the required time to demonstrate survival benefit. PFS is used as an interim measure for OS (surrogate endpoint). The problem in oncology is that this measure does not always correlate well with OS — which is ultimately the most important measure for patients. This is discussed at length in our previous article that was cited in the current work (Meyers et al. 2021). • While we acknowledge the important suggestions from this review regarding categorization of benefits, re-categorizing the data is outside of the scope of the current work and would not change the interpretation of the manuscript, which is, that clinicians overwhelming support all drugs being funded, despite benefitting survival, quality of life or otherwise.
-------------------	---	--

	about how to combine gains in LE and HRQoL. The second way would be to classify drugs in one of four states: (1) Benefits both survival (using hazard ratios or whatever) AND HRQoL; (2) benefits survival, but not HRQoL; (3) benefits HRQoL but not survival; (4) Benefits neither.		
	Similarly, the COIs can be categorized in a loose sequence. First, conflicts with the actual manufacturer are more important than other conflicts. Then, I'd argue, advisory roles are more important than honoraria, conference attendance, etc. The authors may be able to rank the importance of these COIs better than I	Thank you for this comment. We present FCOIs within these subcategories in Table 2. While we offer information about FCOIs, the focus of this manuscript is on the quality of evidence/clinician feedback and	Table 2 Page 17

	can.	therefore out of the scope of our analysis.	
	The analysis would then explore the role of financial conflicts on the various types of drugs, focusing, I'd guess, on the 4th category perhaps – recommending funding when no apparent benefit of any kind existed. I would strongly recommend that the analysis be carried out at the individual recommendation level, not the level of the drug. Table 2 shows the attributes of all 342 individual submissions, but table 3, for reasons I don't understand, has only 48 "recommendations," which (I gather) are the combined recommendation from all physicians responding on each drug.	Thank you for the comment. While we agree with this reviewer that the analysis be carried out on the individual level, it cannot due to CADTH procedures. While clinicians submit individual FCOI documentation, they submit their agreement or disagreement as a group and therefore cannot be disaggregated further. Our analysis is in line with previous research examining CADTH submissions by patient and clinician groups. Lexchin, J. Financial conflicts of interest of clinicians making submissions to the pan-Canadian Oncology Drug Review: a descriptive study. 2019. BMJ Open. We have amended the methods section (analysis) to ensure this is clear.	Data analysis Methods Page 8
	I also think that this analysis could benefit from a multiple regression structure, using models such as logit or probit for the yes/no recommendation choice.	Thank you for this suggestion. We agree, a regression analysis would be suitable to understand which factors that influence the recommendation (yes/no). However, we believe this is outside of the scope of this manuscript. However, we have added this important area of future research to the discussion	Discussion Page 10-11
Reviewer 2	Thank you for this very interesting paper in which you raise a number of relevant	We thank this reviewer for framing the topical nature of	Introduction, Page 5

	points - not least the continued request for high prices for new cancer medicines for solid tumours despite limited health gain (OS and HRQOL - Table 3). This practice has been continuing for some time, e.g. the cost of cancer care also now accounts for up to 30% of total hospital expenditure across Europe (Simoens S et al. What Happens when the Cost of Cancer Care Becomes Unsustainable. European Oncology & Haematology. 2017;13:108-13), with sales of oncology medicines are expected to grow to \$237 billion by 2024 and further with multiple companies researching in this area (Waters R, Urquhart L. EvaluatePharma® World Preview 2019, Outlook to 2024. 2019. - https://info.evaluate.com/rs/607-YGS-364/images/EvaluatePharma_World_Preview_2019.pdf and IMS Institute for Healthcare Informatics. Global Oncology Trend Report - https://www.scribd.com/document/323179495/IMSH-Institute-Global-Oncology-Trend-2015-2020-Report). This is helped by the emotive nature of the disease area (Haycox A. Why Cancer? Pharmacoeconomics. 2016;34:625-7). As a result - see new oncology medicines being launched at high prices with limited health gain - challenging the sustainability of healthcare systems providing universal access for their citizens - discussed in e.g. Cohen D. Cancer drugs: high price, uncertain value. Bmj. 2017;359:j4543 and Godman B et al. Potential approaches for the pricing of cancer medicines across Europe to enhance the sustainability of healthcare systems and the implications. Expert Rev Pharmacoecon Outcomes Res. 2021;21:527-40.	oncology given rising prices and uncertain value. The references provided in this comment are especially useful in the introduction to underline the importance of this manuscript, not only to Canada but also to other countries that include stakeholders in HTA/funding decisions. Added a paragraph to the introduction which includes the current challenges with cancer drugs supported by the references this reviewer provided.	
	a) Introduction - I would move comments on Page 13 lines 33 - 44 plus some of the above to state why Cancer for this paper. The comments you make about Canada are particularly important for other similar healthcare systems - especially the rising costs of cancer	Thank you for this important comment. We have edited the discussion to be less editorial and moved comments to the introduction, including the ones suggested in this comment.	Paragraph added, page 5 Introduction/ Discussion

	medicines in the public fund (which will increase with most medicines to treat patients in ambulatory care now available as low cost generics/ biosimilars)		
	Comments about the value of OS and PFS in solid tumours. There are concerns with how reliable PFS is in solid tumours as a reliable outcome measure especially when resources are constrained (e.g. Paoletti X et al. Assessment of Progression-Free Survival as a Surrogate End Point of Overall Survival in First-Line Treatment of Ovarian Cancer: A Systematic Review and Meta-analysis. JAMA Netw Open. 2020;3:e1918939; Prasad V et al. The Strength of Association Between Surrogate End Points and Survival in Oncology: A Systematic Review of Trial-Level Meta-analyses. JAMA Intern Med. 2015;175:1389-98 and Cortazar P et al. Pathological complete response and long-term clinical benefit in breast cancer: the CTNeoBC pooled analysis. Lancet. 2014;384:164-72 to name just a few). In fact when clinicians, health authorities and pharmacists in the UK established criteria for paying more for new cancer medicines when the purchaser/ provider split was established in the UK - only really concentrated on OS (minimum of 3 to 6 months additional survival) - Ferguson JS et al. New treatments for advanced cancer: an approach to prioritization. Br J Cancer. 2000;83:1268-73. In addition - concerns with some of the leniency of the ESMO criteria among European payers and their advisers - especially regarding the high weighting given to PFS - resulted in the publication of their concerns - Wild C et al. Utilisation of the ESMO-MCBS in practice of HTA. Ann Oncol. 2016;27:2134-6. However - overall endorse the use of ESMO criteria in this paper.	Thank you for this comment which helps understand the nuance of using the ESMO-MCBS scale as a proxy-measure for value. Once again, this Reviewer has provided important references that are incorporated into our limitations. Added limitations of ESMO-MCBS outlined in Wild C et al. into Limitations section.	Limitations Page 12
	It may also be worth commenting that we are seeing an appreciable number of new cancer medicines being launch with	Thank you. This is an important point also. We re-wrote the discussion to be less editorial in	Discussion

	limited data - but still high requested prices (discussed in e.g. Godman et al). Hence do need objective decision making - not unduly influenced by clinicians with a vested interest promoting new cancer medicines at high prices even with limited/ no health gain vs. current standards	nature but included this comment in the first paragraph.	Second paragraph Page 10
Reviewer 3	This paper examine characteristics of clinician input to the pan-Canadian Oncology Drug Review (pCODR) for cancer drug funding recommendations from 2016 to 2020. The motivation for this paper is interesting. It appears to be a survey and statistical report. I have some questions:	Thank you for your review. A comment in response, this article did not employ survey methodology but rather used publicly available data from pCODR/CADTH.	Abstract/Methods describe the source of data
	The authors think that some supported funds may not offer substantial benefits but do not go further into analyzing reasons for the results and how to support them selectively.	We are unclear about what “supported funds” may not offer substantial benefits this reviewer is referring too. Further, we are not sure who this reviewer is referring too to support.	None
	The contribution of the paper is not very clear. How should this information used for fund recommendation be judged for reliability? What information is more helpful in evaluating the potential benefit of funding?	Thank you for this comment. We agree, our recommendations could have been clearer. We have re-written the discussion to include explicit policy implications. Further, implications for the reliability of our findings is underscored in the “strengths and limitations” section.	Strengths and Limitations Page 12
Reviewer: 4	In the article entitled “Association of clinician input and quality of evidence supporting cancer drug funding decisions in Canada” (bmjopen-2022-066378) the authors formulated that they sought to evaluate characteristics of clinician input beyond financial conflict of interests (FCOIs), including the association with funding support and underlying clinical evidence for the cancer drugs under review. The main outcome is clinician	Thank you for the comment. We have also included information related to other countries as these findings relate to all countries with health technology assessment bodies that include a variety of stakeholders in funding decisions.	Introduction Paragraph 1 Page 5 & Discussion

	support for the initial funding recommendation. The paper is fairly well written and primarily written for the Canadian health system.		
	In the strengths and limitations the authors claim that the study is the first to examine the association between clinician funding recommendations and the evidence underpinning cancer drug reimbursement decisions. This claim should be clearly substantiated and argued with respect to Lexchin's publication (3) and Meyers' two publications (1,2) given the period under investigation and the variables used to justify this claim.	Thank you for this comment. We meant that, to our knowledge, this study is the first to examine clinician input in relation to the quality of evidence that underpins the drugs. We have altered the language to ensure clarity.	Strengths and Limitations Page 3
	Table 5 of Lexchin's article also examines an association. In Lexchin's article this association is analysed using a Fisher exact test, which test was not used in the bmjopen-2022-066378, where the associations are only presented descriptively. So the addition of, say, a Fisher exact test or other tests would already be an improvement to the associations presented in table 3 (page 11).	Thank you for this comment. In line with the Editors comment, we removed the reference to "association" throughout the manuscript given the descriptive focus of the manuscript. While we could employ a Fishers Exact test in Table 3, we feel that this method would not add to the paper as the trend is visible without the use of a statistical test. Therefore, we would rather not given its potential inappropriateness with numerous "0" cell counts.	The use of "association" was removed throughout manuscript.
	Other research techniques that could be applied in this manuscript under review are network analysis, but multilevel analysis could also be a useful analysis technique to analyse the information currently scattered in Tables 1, 2 and 3 into a single whole. This allows covariates to be taken into account in a multilevel analysis.	Thank you for this comment. We agree that there are sophisticated statistical techniques that would be interesting to use within this context. In line with the previous reviewer comment about the use of logit/probit regression, we have added this point in the Discussion for future research.	Discussion Page 10-11
	The figure on page 18 could be better presented by placing the 100% conflict submission numbers together. The submission numbers have no meaning to the reader. Moreover, the numbers 29 and 35 are disturbing and should be	Thank you. The figure has been adapted with this reviewer's advice.	Figure.

	dropped from the figure and can be listed at the bottom of the table as has been done now.		
	The study does not use any inferential statistics to examine associations between clinician funding recommendation and supporting evidence. Only descriptive statistics are presented. It seems to be difficult to compare proportions between two groups due to that a number of clinician recommendation with “do not fund” is too small (n=2); any associations may be found by chance. Although Chi-square test is not applicable, Fisher exact test is suggested to use for examining associations between the variables.	Thank you for this comment. In line with the Editors comment, we removed the reference to “association” throughout the manuscript given the descriptive focus of the manuscript. While we could employ a Fishers Exact test in Table 3, we feel that this method would not add to the paper as the trend is visible without the use of a statistical test.	The use of “association” was removed throughout manuscript.
	In tables, numbers and percentages of the study samples should be provided in all cells and rechecked. Moreover, there is no analysis section in this study. It should be described and informed audience how this study treated and analyzed the data in particular to examine the associations.	Thank you for this comment. We have rechecked all the data presented in this manuscript. Further, we add clarifying language in the methods about the proportions analysed. We also added a “Data analysis” section in the Methods section.	The use of “association” was removed throughout manuscript. Data Analysis section Page 8
	Study design is defined as a retrospective cohort study. I think that it can be named a retrospective study but do not think that a cohort study is applicable. However, if a terms cohort study is used, some epidemiological terms should be clearly defined i.e. exposure and non-exposure, outcomes measurement, incidence and follow-up time.	Thank you for this comment. While a group of drug submissions can be considered as a “cohort” we agree that it might be confusing. We have changed the study design to “cross-sectional, descriptive study”.	Abstract, Methods Page 2

VERSION 2 – REVIEW

REVIEWER	Phelps, Charles University of Rochester
REVIEW RETURNED	25-Jun-2023

GENERAL COMMENTS	In this revision, the authors have still made no attempt to link FCOIs with recommendations. This leaves the study with a somewhat uninteresting conclusion, namely that Canadian clinicians will recommend funding of almost all cancer drugs, no matter what their clinical benefit. The one policy recommendation that might come of this would be to save the costs of having the clinician input, since clinicians seem to support drug funding independent of the clinical gains. It would appear to be possible to illuminate the role of FCOIs by doing a simple cross-tab table of support for the drugs by outcome status vs. FCOI status. This would simply test whether there were statistically significant differences in FCOI status separately for drugs that did and did not have OS benefit, and similarly for PFS survival. The key test would be positive recommendations for drugs that offered no gains in any area – OS, PFS and HRQoL. Does FCOI status differ here? I cannot understand why such analyses have not been undertaken here. In my prior review, I urged this sort of two-way analysis, but this manuscript basically continues to have only one-way analyses. I would reject it or return for a massive revision. I would not wish review another round unless the authors carried out this sort of analysis. The one-way tables have little interest.
---

REVIEWER	Stewart, Roy Rijksuniversiteit Groningen, Department of Health Sciences, Community and Occupational Medicine, University Medical Center Groningen
REVIEW RETURNED	21-Jun-2023

GENERAL COMMENTS	First of all, I would like to thank the auteurs for their reply and the changes in the manuscript. From my point of view, the paper has improved. However, there is still a question (also asked by another reviewer) that has not yet been answered and there is another comment on my behalf:  1. In the authors' response, it is stated that the trend is visible without the use of a statistical test. But nowhere is the trend discussed properly. So the question is where is the trend discussed? The authors should explain this better and more because no statistical analysis is done. 2. The discussion vaguely formulates the so-called sophisticated analyses. It would be better to properly formulate which analysis techniques can be used in future research.
---

VERSION 2 – AUTHOR RESPONSE

Reviewer: 4
Dr. Roy Stewart, Rijksuniversiteit Groningen

Comments to the Author:

First of all, I would like to thank the auteurs for their reply and the changes in the manuscript. From my point of view, the paper has improved. However, there is still a question (also asked by another reviewer) that has not yet been answered and there is another comment on my behalf:

1. In the authors' response, it is stated that the trend is visible without the use of a statistical test. But nowhere is the trend discussed properly. So the question is where is the trend discussed? The authors should explain this better and more because no statistical analysis is done.

Thank you for this comment. Originally, we were discussing the trend that physicians nearly always want the drug funded (46 of 48 submission, 96%) vs not funded (2 of 48 submissions, 4%). However, we agree with both reviewers that this led to a rather uninteresting conclusion and therefore decided to extend our analysis to the association between FCOIs and funding recommendations (Table 4, page 19) and the association between clinician funding recommendation for drugs with or without health benefits (Table 5, page 20).

The changes and details of this analysis are provided in response to Reviewer #2 (comment #2)

2. The discussion vaguely formulates the so-called sophisticated analyses. It would be better to properly formulate which analysis techniques can be used in future research.

We agree that the discussion did not support the initial analysis. As discussed above, we have added in further analyses. The details and changes to the manuscript are provided in response to Reviewer #2 (comment #2). Additionally, we added suggestions for future research:

“Future research might build on our study to further examine the effects of FCOIs on the CADTH decision with more sophisticated statistical methods or qualitative analysis.” (Page 13; line 365-367).”

And tempered our findings throughout the discussion:

“However, there was no statistically significant differences between the distribution of individual clinician FCOIs and recommendation for drugs with meaningful clinical benefit. This is an interesting finding that warrants further investigation to understand whether a true cause exists but may be related to the previously discussed desire to have numerous options for patients given treatment responses that may differ from the clinical trial results. Given the overwhelming support for nearly all drugs in our sample, this finding may be skewed. Future research might repeat this study with the inclusion of haematological malignancies given the recently released ESMO-MCBS for this population.” (Page 11-12; line 307-315).

Reviewer: 1

Dr. Charles Phelps, University of Rochester

Comments to the Author:

In this revision, the authors have still made no attempt to link FCOIs with recommendations. This

leaves the study with a somewhat uninteresting conclusion, namely that Canadian clinicians will recommend funding of almost all cancer drugs, no matter what their clinical benefit. The one policy recommendation that might come of this would be to save the costs of having the clinician input, since clinicians seem to support drug funding independent of the clinical gains.

Thank you for taking the time to review our study. We agree with this conclusion and have added further analyses. We discuss this in the next comment where this reviewer has provided helpful recommendations for the analysis. Furthermore, we have added the policy recommendation to the discussion (page 12, line 339 to 344 in the Clean Copy).

“To manage provider-induced, CADTH may want to reconsider how the agency engages with clinicians, or whether the agency ought to incorporate external stakeholder feedback on every funding decision, given overwhelming support for funding nearly every drug independent of clinical benefit . Furthermore, CADTH, among other HTA bodies, may want to be more explicit for how stakeholder input is used in funding decisions.”

It would appear to be possible to illuminate the role of FCOIs by doing a simple cross-tab table of support for the drugs by outcome status vs. FCOI status. This would simply test whether there were statistically significant differences in FCOI status separately for drugs that did and did not have OS benefit, and similarly for PFS survival. The key test would be positive recommendations for drugs that offered no gains in any area – OS, PFS and HRQoL. Does FCOI status differ here? I cannot understand why such analyses have not been undertaken here.

Thank you for the helpful suggestions for this analysis. We agree, the data are available and would be important to include. For this reason, we have added two tables exploring the associations between clinician group recommendations and conflicts of interest, including one table aggregated by clinical gains, as suggested by this reviewer.

- First, we explore FCOI status and group recommendations to fund or not to fund the drug. We find that the distribution among the categories do differ statistically. However, we note in our manuscript that our sample is heavily skewed towards positive funding recommendations and this analysis can be repeated with a larger sample (especially now that the ESMO-MCBS tool is in development for haematological malignancies) (Page 11-12; line 307-315).
- Table 4 (page 20) provides an overview of the role of FCOIs per meaningful health gains (defined as OS, ESMO-MCBS tool, and HRQoL) for drugs that received positive clinical group recommendations. We found no significant difference between the FCOI status and clinical benefit, which suggests clinicians seem to support funding all drugs independent of health gains.

We have added details of this analysis in the Methods, Results, and Discussion sections where appropriate.

Reviewer: 4

Competing interests of Reviewer: None

Reviewer: 1

Competing interests of Reviewer: None

VERSION 3 – REVIEW

REVIEWER	Stewart, Roy Rijksuniversiteit Groningen, Department of Health Sciences, Community and Occupational Medicine, University Medical Center Groningen
REVIEW RETURNED	20-Aug-2023

GENERAL COMMENTS	I would like to express my thanks for being given the opportunity to review this manuscript and also my thanks to the authors for answering my questions. As an interested reader, you try to establish relationships between the tables, but it is not always clear e.g. Clinical gains are mentioned in table 4 and they are also found in table 2, but how the numbers between these two tables relate is not clear. I therefore have the impression that the data presented in the different tables are not in one dataset, because if all data are in one dataset, then it is possible to taking into account for other variables in the associations presented in Table 4 (Association between the number of clinicians declaring financial conflicts of interests and supporting evidence for positive group recommendations). Should it be the case that aggregate data is analysed, this may be seen as a limitation. A few minor revision: 1. Line 16 says that there are 5 tables, but I only see 4.2. There should be an addition of the percentage in line 227 at the 'overwhelming majority of clinicians' who were in favour of funding the drug under review.3. Line 259 mentions "(Table)", but this needs to be filled in.4. I wonder whether the page number entered in the form "STOBE Statement – Checklist of items that should be included in reports of cohort studies" is still correct.
---

VERSION 3 – AUTHOR RESPONSE

Comments to the Author:

I would like to express my thanks for being given the opportunity to review this manuscript and also my thanks to the authors for answering my questions.

As an interested reader, you try to establish relationships between the tables, but it is not always clear e.g. Clinical gains are mentioned in table 4 and they are also found in table 2, but how the numbers between these two tables relate is not clear. I therefore have the impression that the data presented in the different tables are not in one dataset, because if all data are in one dataset, then it is possible to taking into account for other variables in the associations presented in Table 4 (Association between the number of clinicians declaring financial conflicts of interests and supporting evidence for positive group recommendations). Should it be the case that aggregate data is analysed, this may be seen as a limitation .

We thank this reviewer for taking the time to read our manuscript and provide thoughtful comments that improve its clarity.

1. We have added further information about the data to the methods section that specify where the data come from and how they were used for this analysis. Page 8 (194-196)
2. We added the Reviewer's point about the aggregate data to the manuscript: "Data per submission were aggregated to compare these factors" (Page 208-09)
3. The nature of the data is included as limitation in the manuscript (page 12; 356-60)

A few minor revision:

1. Line 16 says that there are 5 tables, but I only see 4.

- Thank you, we initially provided another table, but this was over the BMJ limit. Therefore, we have changed the tables to 4.

2. There should be an addition of the percentage in line 227 at the 'overwhelming majority of clinicians' who were in favour of funding the drug under review.

- Thank you for this comment. This sentence has been deleted as the following sentence describes the actual proportion of submissions with clinician group feedback that were in favor of funding the drugs.

3. Line 259 mentions "(Table)", but this needs to be filled in.

Thank you. This has been changed to "(Table 4)".

4. I wonder whether the page number entered in the form "STOBE Statement – Checklist of items that should be included in reports of cohort studies" is still correct.

Thank you. This reviewer is quite right. STROBE Statement needed to be updated after the major revisions. A new document has been uploaded.